

# Clinical implications of aberrant neurovascular structures coursing through the submandibular gland

Kelsey J. Eaton[1] and Heather F. Smith[2,3]

[1] Department of Osteopathic Manipulative Medicine, Midwestern University, Glendale, AZ, United States of America
[2] Department of Anatomy, Midwestern University, Glendale, AZ, United States of America
[3] School of Human Evolution and Social Change, Arizona State University, Tempe, AZ, United States of America

## ABSTRACT

**Background**. Variation within the submandibular triangle, including variant paths of facial neurovasculature, could increase risk of neurovascular derangement during submandibular gland (SMG) dysfunction, enlargement, interventions, or removal.
**Methods**. Frequency of anatomical variants enveloped within or piercing the SMG, including facial artery, vein, or branches of CN VII, were assessed in 70 cadaveric submandibular glands (39M/31F).
**Results**. Eighteen of 70 SMGs (25.7%) were pierced by at least one aberrant neurovasculature structure: Facial artery most frequently ($n = 13$), followed by facial vein ($n = 2$), inferior labial artery and vein ($n = 1$), and CN VII cervical branch ($n = 1$). This study demonstrated the high variability of neurovasculature within submandibular parenchyma. These aberrant neurovascular structures, especially facial artery, are in danger of compromise during surgical and other medical procedures on the SMG. To avoid potential neurovascular compromise, ultrasonographic or other imaging is recommended prior to procedures involving the SMG.

## INTRODUCTION

The submandibular triangle contains a complex interaction of anatomy and physiology, allowing for numerous health conditions and procedures to occur. This well-defined region is outlined by the inferior border of the mandible and the anterior and posterior bellies of the digastric muscle (Fig. 1). The superficial and deep boundaries are limited by the platysma and the mylohyoid muscle, respectively, creating a limited space for its many contents. The submandibular gland (SMG), also known as the submaxillary gland, typically occupies the majority of the submandibular triangle.

The SMG is vulnerable to eclectic pathology that involves both clinical and surgical interventions (*Wilson, Meier & Ward, 2014*; *McQuone, 1999*; *Iro & Zenk, 2014*; *Schiødt et al., 1992*). Enlargement of the gland may occur, most commonly due to salivary flow obstruction, infection, autoimmune disorders, and iatrogenic consequences. The current

Corresponding author
Heather F. Smith,
hsmith@midwestern.edu,
Heather.F.Smith@asu.edu

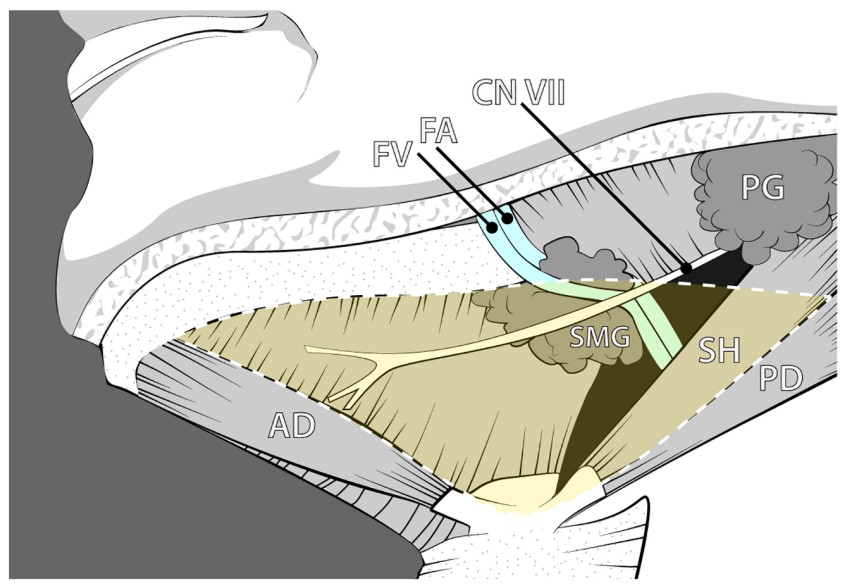

**Figure 1** **Anatomy of the submandibular region illustrating the classic anatomical relationships of the submandibular gland and associated neurovasculature.** Abbreviations: AD, Anterior digastric; CN VII, Facial nerve; FA, Facial artery; FV, Facial vein; PD, Posterior digastric; PG, Parotid gland; SH, Stylohyoid; SMG, Submandibular gland. Drawing source credit: Brent Adrian.

standard of care for enlargement suspicious for neoplastic origin is Fine Needle Aspiration (FNA). This procedure samples the gland contents; however, its findings may result in indefinite answers due to the nature of the sampling technique as FNA has varying reported rates of sensitivity (*To et al., 2012*). While neoplastic occurrence is relatively rare in the SMG, positive diagnostic findings typically result in removal of the gland. Thus, for patients with positive or inconclusive neoplastic testing, the treatment plan begins with removal of the gland (*Agni & Borle, 2013*; *Smith, Peters & Markus, 1993*; *Rapidis et al., 2004*). Patients with neurological deficits such as cerebral palsy can similarly be affected by iatrogenic changes to the SMG. Those who suffer from sialorrhea, an overproduction of saliva, may undergo Botox injection to minimize saliva production, ligation of the submandibular salivary duct, or even complete SMG removal to help alleviate refractory symptoms (*Manrique & Sato, 2009*). Finally, the SMG's shape or position can be altered for aesthetic correction from SMG ptosis (*Sullivan, Freeman & Schmidt, 2006*).

The pathologies and procedures of the submandibular triangle occur with varying frequency; however, care providers must be aware of the restrictions and variances within this space. Accurate knowledge of the contents and surroundings of the SMG will facilitate designing appropriate surgical planning, and possible explanation, prevention, or treatment of sequelae following structural derangement within the submandibular triangle.

Given the relatively small size of the submandibular anatomical space and complex multi-staged developmental pattern, we hypothesize that there is higher likelihood of significant variation of the neurovascular pathways associated with the SMG than previously reported. Classically, vascular and neural considerations placed the facial artery within a posterior

path along an open ceiling canal of the SMG (Fig. 1). However, aberrant pathways in which the facial artery may deviate and course within the parenchyma of the gland could put the vasculature and nerves at risk in cases of pathology, surgical procedures, or other derangement of the submandibular gland. In fact, a few cases of the facial artery piercing the SMG have been reported in the literature (*Vadgaonkar et al., 2012*; *Venugopal et al., 2014*). The integrity of the facial artery is also relevant given the postsynaptic sympathetic periarterial plexus that travels on its external surface. These concerns are in addition to the more well-known complications of the marginal mandibular branch of facial nerve or other lower facial nerve branches that are at risk of iatrogenic damage that are due to close proximity to the SMG (*Ichimura, Nibu & Tanaka, 1997*; *Bates et al., 1998*).

## MATERIALS & METHODS

Seventy cadaveric submandibular glands (39M/31F) from the gross anatomy teaching laboratories at Midwestern University were assessed to determine the frequency of variance in submandibular gland neurovasculature within the submandibular triangle (Fig. 1). Following student dissection, the integrity and presence of the SMG was first evaluated, as the student coursework required dissection through the gland or complete removal in some cases. Cadaver inclusion depended on a minimum of unilateral presence of the gland. Additional dissection and preparation were necessary to further reveal the SMG and its associated neurovascular structures in several cadavers. Careful dissection was conducted to assess whether any neurovascular structures were coursing through the gland tissue, as opposed to coursing posteriorly and adhering to the deep surface of the gland.

Data were then collected on the presence and frequency of neurovasculature enveloped within the SMG. Neurovascular structures excluded from consideration were minor branches directly off the facial artery and the marginal mandibular nerve of CN VII, as these are considered known and accepted variants. For each gland, data on sex and laterality of variants were also recorded. Chi-squared statistical analyses were conducted in SPSS 19 (IBM Corp.) to determine whether significant differences existed in frequency of variants between sexes or sides of the body. This study was determined to be IRB-exempt by the Midwestern University Institutional Review Board, due to the subjects being entirely cadaveric (Exemption #AZ1121).

## RESULTS

In the seventy preserved SMGs, 18 submandibular glands (25.7%) were found to demonstrate variations from the classical anatomy (Table 1). One of these glands contained two variations within it, allowing for a total of 19 variant structures (Table 1). Variations were defined as neurovasculature not classically associated with direct gland innervation or blood supply. Most variants were found to be present unilaterally, with only one female cadaver possessing bilateral variation.

The predominant variation observed was the facial artery encapsulated by the SMG, rather than traveling posteriorly, which comprised 68.4% of the observed variations ($n = 13$) totaling 18.6% of all SMGs (Table 2; Fig. 2). This variant was followed in frequency

**Table 1  Frequency of anatomically variant neurovascular structures within 18 of the 70 submandibular glands inspected.** Total variant structures were significantly more common in females than in males ($\chi^2 = 4.92$, $p = 0.027$).

| Category | Facial artery | Facial vein | Inferior labial artery[a] | CN VII cervical branch | Unidentified neurovasculature | Total variant structures |
|---|---|---|---|---|---|---|
| Male | 4 | 1 | 0 | 1 | 1 | 7 |
| Female | 9[b] | 1 | 1 | 0 | 1 | 12 |
| Left | 8 | 2 | 0 | 1 | 0 | 11 |
| Right | 5 | 0 | 1 | 0 | 2 | 8 |
| **Total** | **13** | **2** | **1** | **1** | **2** | **19** |

Notes.
[a] Inferior labial artery branched proximally, through the submandibular gland.
[b] One female cadaver had bilateral involvement of the facial artery in both SMGs.

**Table 2  Sample size, laterality, and sex of cadaveric subjects with the most common anatomical variant of the submandibular gland, the facial artery piercing the SMG.** The facial artery pierced the SMG in significantly higher frequency in females compared to males ($\chi^2 = 4.03$, $p = 0.045$).

| Category | Number of SMGs | Facial artery variants |
|---|---|---|
| Male | 39 | 4 (10.3%) |
| Female | 31 | 9[a] (29.0%) |
| Left | 38 | 8 (21.1%) |
| Right | 32 | 5 (15.6%) |
| **Total** | **70** | **13 (18.6%)** |

Notes.
SMG, Submandibular Gland
[a] One female cadaver displayed bilateral involvement of the SMGs.

by the facial vein ($n = 2$), inferior labial artery and vein ($n = 1$), and cervical branch of CN VII ($n = 1$) piercing the gland (Table 2). There were two additional unidentified neurovascular bundles observed within gland parenchyma that had been severed from previous dissection. These unidentified bundles were traced proximally and confirmed as being neither the glandular branches from the facial artery nor the marginal mandibular nerve, which were both intact.

Chi-squared tests revealed significant sex differences in the frequency of neurovascular anomalies. Female subjects displayed a significantly higher overall rate of piercing structures compared to males ($12/31 = 38.7\%$ and $6/39 = 15.4\%$, respectively), $\chi^2 = 4.92$, $p = 0.027$. Females also demonstrated a significantly higher rate of the facial artery piercing the SMG ($9/33 = 27.3\%$) than males ($4/39 = 10.3\%$) with $\chi^2 = 4.03$, $p = 0.045$. Comparisons of the frequency of variants between the left and right sides were insignificant for overall variants ($\chi^2 = 0.14$, $p = 0.71$), and for the facial artery ($\chi^2 = 0.34$, $p = 0.56$). The low frequencies in the other neurovascular variants yielded sample sizes that were insufficient for statistical comparisons.

## DISCUSSION

The previously accepted anatomical pattern of the submandibular triangle presumed that the facial artery courses via a posterior canal alongside the SMG as it provided blood supply

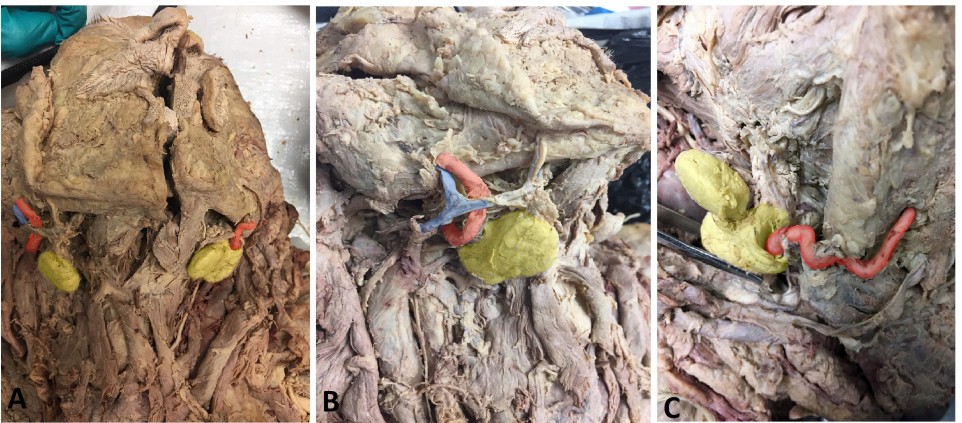

**Figure 2** **Photos demonstrating anatomical variations in neurovasculature around the submandibular gland.** (A) Anterior view of a hemisected cadaveric specimen demonstrating both classic (left side of photo) and variant (right side of photo) anatomy; (B) Right SMG showing the classic anatomical condition in which no neurovasculature pierces the SMG; (C) Left SMG showing a variant condition in which the facial artery courses through the SMG. Photo has been colored for easy identification of structures: Yellow, Submandibular gland; Red, Facial artery; Blue, Facial vein.

via glandular branches (*Sidell & Shapiro, 2012*). A prevalence of variation in this arterial path has not been widely established. The high frequency of aberrant anatomy revealed in this study (25.7%) indicates that there is greater variation in the submandibular triangle than commonly acknowledged. Additionally, we report for the first time that such variants are significantly more common in females than in males.

Surgical procedures are the most obvious clinical application of these variant patterns; however, many pathologies causing glandular enlargement are of both surgical and clinical interest. These pathologies range from disorders such as neoplasms, diabetes, hypothyroidism, autoimmune disorders, and viral infections (*McQuone, 1999*; *Iro & Zenk, 2014*; *Schiødt et al., 1992*; *Smith, Peters & Markus, 1993*). When SMG enlargement occurs in patients with variant neurovasculature compression by the gland may lead to theoretical nerve entrapment, salivary consistency and production changes, and/or intraoperative complications dependent on the structure enveloped (*Ximenes Filho, Imamura & Sennes, 2002*). Therefore, structural knowledge is essential in understanding each patient's true functionality and pathology, especially those presenting with atypical symptoms or those refractory to treatment. Clinically, the neurological structure previously considered most at risk during common procedures involving the SMG was the marginal mandibular nerve (CN VII) as it lies adjacent to the SMG with utilization of the facial vein and artery as landmarks for preservation (*Ichimura, Nibu & Tanaka, 1997*; *Bates et al., 1998*). While the risk to the marginal mandibular nerve during surgery is well documented, we believe that the above results provide sufficient evidence of other neurovascular structures at risk that must be considered prior to interventions in this region.

Anatomically, knowledge of neural and vascular location is pivotal within surgical specialties. The facial artery may be readily sacrificed in operations if necessary, for

advancement, visualization, or mobilization (*Holsinger & Bui, 2007*). Due to the amount of collateral blood flow, removal of the facial artery has no clinical consequence outside of hematoma risk from ligation failure or infections (*Mendelson & Tutino, 2015*). However, knowledge of this piercing variant is important after trauma or locally invasive neoplasm as the facial artery may be used for facial flaps in follow-up reconstructive procedures (*Talmi et al., 2003*).

Other procedural considerations are those within field of aesthetic correction. As fascia and tissues become increasingly lax with age, the submandibular gland can experience ptosis, creating undesired enlargement under the mandible (*Sullivan, Freeman & Schmidt, 2006*; *Mendelson & Tutino, 2015*). Several procedures are of note, including SMG suspensions and shaving. The former cosmetic procedure elevates the glands from below the mandible providing a rejuvenating appearance and does not routinely involve ligation of the facial artery since the SMG is preserved (*Sullivan, Freeman & Schmidt, 2006*). A shaving technique with similar results involves surgical contouring of the SMG, which includes explicit concerns over the location of the facial artery and its penetrating branches (*Mendelson & Tutino, 2015*).

Clinically, we postulate several reasonable consequences or potential complications due to these newly identified anatomical variants in the submandibular triangle. Case reports have shown pseudoaneurysm occurrence with botulinum (Botox) injection or from trauma to vessels of the neck and face (*Dediol et al., 2011*; *Ribeiro-Ribeiro, Junior & Pinheiro, 2011*). The procedure is associated with varying commitment to ultrasonography of the gland's position and structure prior to injection which may lead to inaccurate placement of the needle (*So et al., 2017*). Knowledge of these anatomical variants and the possibility of pseudoaneurysm and hematoma risk predicates the necessity of consistently using ultrasound prior to injection so as to avoid neural or vascular structures. This may be especially true with many of the populations that require SMG interventions as they may have already poor nutritional status or previous radiotherapies, which could predicate pseudoaneurysm formation (*Patel, Patterson & Chapple, 2006*).

Salivary tumors probability of malignancy changes in relation to the gland. The accepted rule of thumb is that with decreasing size of the gland, there is an increased incidence of malignancy (*Agni & Borle, 2013*). This is demonstrated by the 25/50/75 rule for parotid, submandibular, and sublingual glands respectively (*Agni & Borle, 2013*). Dependent on tumor subtype salivary gland neoplasms may spread preferentially either using lymphangiogenesis or hematogenously with distant metastasis (*Hertzer, 2000*). A neoplasm arising in an SMG that contains a facial artery, vein, or other aberrant structure could have additional consequences and considerations such as metastasis risk or compression. Arterial walls are more resistant to infiltration or compression by neoplastic cells in comparison to the thinner-walled veins. However, as mentioned previously, the facial artery transports a periarterial postganglionic sympathetic nerve plexus on its external walls, which would be at risk. Certain neoplasms such as adenoid cystic carcinomas, the most common location being in the minor and major salivary glands, can spread along these nerves in a perineural manner well before it is visible on imaging (*Fujita et al., 2011*; *Vrielinck et al., 1988*).

Postganglionic sympathetic nerves may be at risk in non-malignant conditions as well. Stricture, enlargement, or fibrosis such as in chronic sclerosing sialadenitis (Kuttner's Tumor) could reasonably apply pressure to the sympathetic plexus over the facial artery. This relationship may play a role in salivary dysfunction and warrants further study as the glands receive input from both the sympathetic and parasympathetic systems (*Jaso & Malhotra, 2011*; *Proctor & Carpenter, 2007*). Unlike much of the rest of the body in which these systems have antagonistic effects, in the salivary glands, both sympathetics and parasympathetics stimulate production of saliva. The difference between the input of the two systems results in volume and consistency changes (*Delfs & Emmelin, 1974*).

For all procedures and interventions outlined above, an understanding of the neurovascular anomalies reported here would allow for appropriate planning and risk-benefit consideration. Being aware of the involvement of nerves and arteries coursing through the gland can facilitate the improvement of informed consents when discussing the procedure and its potential complications. In order to better understand the prevalence of aberrant anatomy within the submandibular region, further study could focus on a sample of live subjects using ultrasonography to assess rates of anatomical variance.

## CONCLUSIONS

This study revealed a high degree of variability in the neurovasculature within the submandibular triangle, identifying the high frequency of anomalous facial artery patterns. As a result, compression of these structures should be considered when treating a patient with lower head and neck symptomatology. The data also indicate a statistically higher rate of aberrant submandibular neurovasculature in females than in males. Knowledge of possible variance allows for individualized and comprehensive treatment plans, more informed surgical approach, and facilitation of more targeted procedures when necessary.

## ACKNOWLEDGEMENTS

The authors would like to thank the generous body donors whose cadavers formed the basis of this study. Figure 1 was created by Brent Adrian. Thank you to Ashley Bergeron and the Anatomical Laboratories staff for their accommodation in the anatomy laboratories. We thank Drs. Robert Druzinsky, David Reed, and an anonymous reviewer for constructive feedback that improved the paper.

### Funding

This work was funded by Midwestern University. The funders had no role in study design, data collection and analysis, decision to publish, or preparation of the manuscript.

### Grant Disclosures

The following grant information was disclosed by the authors:
Midwestern University.

## Competing Interests

The authors declare there are no competing interests.

## Author Contributions

- Kelsey J. Eaton conceived and designed the experiments, performed the experiments, analyzed the data, prepared figures and/or tables, authored or reviewed drafts of the paper, approved the final draft.
- Heather F. Smith conceived and designed the experiments, analyzed the data, contributed reagents/materials/analysis tools, prepared figures and/or tables, authored or reviewed drafts of the paper, approved the final draft.

## Human Ethics

The following information was supplied relating to ethical approvals (i.e., approving body and any reference numbers):

This study was determined to be IRB-exempt by the Midwestern University Institutional Review Board, due to the subjects being entirely cadaveric (Exemption #AZ1121).

## Data Availability

The data file of anatomical variants noted for each specimen is available in the Supplemental File.

## Supplemental Information

Supplemental information for this article can be found online at http://dx.doi.org/10.7717/peerj.7823#supplemental-information.

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
