# Peer review of "Clinical implications of aberrant neurovascular structures coursing through the submandibular gland"

_PeerJ, doi:10.7717/peerj.7823_

## Round 0.1 · original submission · Minor Revisions

Both reviews liked your manuscript very much. Please address the minor issues raised by the reviewers to make it even better and re-submit it.

·

Basic reporting

While it is acknowledged in the head and neck clinical literature that the facial artery sometimes passes through the SMG, I was surprised to find that there were no percentages of this variant reported in the literature. This manuscript will be a valuable contribution to any clinical study interested in reporting anatomic variations in the submandibular triangle.

I did find two papers that discuss variations in the submandibular triangle that are not cited in this manuscript. Please consider including these papers in their revision:

Vadgaonkar, Rajanigandha, et al. "Variant facial artery in the submandibular region." Journal of Craniofacial Surgery 23.4 (2012): e355-e357.

Venugopal, S. V., et al. "RELATIONSHIP BETWEEN THE FACIAL ARTERY AND SUB MANDIBULAR SALIVARY GLAND." Int J Anat Res 2.3 (2014): 597-600.

The data are clear and unambiguous

The manuscript is well written. There are a couple minor typos that the authors should consider fixing
ln 35 "for numerous health conditions and procedures to be occur"
Edit: Wording
Ln 44 "for neoplastic origin, is Fine Needle Aspiration"
Edit: comma

Experimental design

no comment

Validity of the findings

no comment

Additional comments

The authors have done an excellent job with the discussion. It was thorough and highly relevant. This manuscript will be a great resource.

Reviewer 2 ·

Basic reporting

The language of the manuscript was in clear and professional English. Proper background and context was provided and citations were appropriate. The figures and tables were very well done and offered clear detail. There were a few instances where the flow of the text wandered off course or was redundant (e.g., line 48, “Thus…”), but his was not really detrimental to the text.

Experimental design

The research question was well defined and meaningful. The relevance of the investigation is, to me, not of critical importance as surgical variations are well known and surgeons and anatomists have been working in this region for generations. The important aspect of this manuscript is the frequency data of each structure found within the substance of the submandibular gland. The research was conducted on body donors from a medical school anatomy lab (with documentation of IRB waiver), which introduced some limitations to the study, which was clearly addressed in the text.

Validity of the findings

My only concern is the possibility of some bias due to the inclusion of donors with a minimum of one gland. If a donor with a damaged gland that may or may not have had a variation is compared with both grands from another donor, some bias might creep into the dataset and influence the overall results. Also, using a chi-square test that is robust to the effect of small sample size is likely appropriate. Other than those thoughts, the data were collected and documented well, appropriately defined, and support the overall conclusions of the manuscript.

---

## Round 0.2 · accepted · Accept

Thank you for your timely re-submission and consideration of each of the comments by the two reviewers.